# Enhanced optical path and electron diffusion length enable high-efficiency perovskite tandems

Bin Chen[1,3], Se-Woong Baek[1,3], Yi Hou [1,3], Erkan Aydin[2], Michele De Bastiani[2], Benjamin Scheffel[1], Andrew Proppe [1], Ziru Huang [1], Mingyang Wei [1], Ya-Kun Wang [1], Eui-Hyuk Jung[1], Thomas G. Allen[2], Emmanuel Van Kerschaver[2], F. Pelayo García de Arquer [1], Makhsud I. Saidaminov[1], Sjoerd Hoogland[1], Stefaan De Wolf [2] & Edward H. Sargent [1✉]

Tandem solar cells involving metal-halide perovskite subcells offer routes to power conversion efficiencies (PCEs) that exceed the single-junction limit; however, reported PCE values for tandems have so far lain below their potential due to inefficient photon harvesting. Here we increase the optical path length in perovskite films by preserving smooth morphology while increasing thickness using a method we term boosted solvent extraction. Carrier collection in these films – as made – is limited by an insufficient electron diffusion length; however, we further find that adding a Lewis base reduces the trap density and enhances the electron-diffusion length to 2.3 μm, enabling a 19% PCE for 1.63 eV semi-transparent perovskite cells having an average near-infrared transmittance of 85%. The perovskite top cell combined with solution-processed colloidal quantum dot:organic hybrid bottom cell leads to a PCE of 24%; while coupling the perovskite cell with a silicon bottom cell yields a PCE of 28.2%.

[1] Department of Electrical and Computer Engineering, University of Toronto, 35 St George Street, Toronto, ON M5S 1A4, Canada. [2] KAUST Solar Center (KSC), Physical Sciences and Engineering Division (PSE), King Abdullah University of Science and Technology (KAUST), Thuwal 23955-6900, Kingdom of Saudi Arabia. [3]These authors contributed equally: Bin Chen, Se-Woong Baek, Yi Hou. ✉email: ted.sargent@utoronto.ca

andem solar cells consisting of multiple absorber layers with different bandgaps reduce thermalization losses and offer a further increase in the power conversion efficiency (PCE) of single-junction cells[1]. For perovskite solar cells (PSCs), single-junction device PCEs have increased from 3.8 to 25.2% over the course of a decade[2,3]. As these devices usually employ materials having a bandgap of 1.5–1.7 eV, they are also particularly attractive as top cells in tandem devices. They can be coupled with perovskite[4,5], organic[6], colloidal quantum dot (CQD)[7–9], crystalline silicon[10,11], and copper indium gallium selenide (CIGS)[12,13] solar cells in both two-terminal (2T) and four-terminal (4T) configurations. The 4T tandem arrangement offers a broader bandgap selection window for the constituent cells[1,14]. Perovskite-silicon 4T tandem solar cells recently reported in the literature have reached an impressive PCE of 27.1%[15].

Efficient harvesting of the photons above the bandgap of each subcell is needed to realize the potential of the tandem concept[16]. To this end, both front and back electrodes of the top cell need to be highly transparent to allow near-infrared (NIR) photons to reach the bottom cell; and such a configuration reduces the internal reflection of photons close to the perovskite bandedge, making the achievement of high absorption of above-bandgap photons in the top cell challenging—for this must be accomplished in a single pass through the active layer. In general, the external quantum efficiency (EQE) of efficient single-junction perovskite cells with opaque rear-mirror contacts (metal) is above 80% near the bandedge[17–19], while that of semi-transparent cells show a lower value (around 70%) in the same spectral range[4,13,20–22]. Employing high-quality and thick active layers to overcome insufficient photon absorption has advanced Sn-based perovskite solar cells[4,5,23]; however, for Pb-based perovskites, thick films exhibit inhomogeneous morphologies[24,25] and limited carrier diffusion lengths[26–28].

Here, we find that the formation of a wrinkled surface in thick perovskite films is due to the high precursor concentration used in the conventional solution processing method. We develop a technique, which we term boosted solvent extraction, that enables a homogeneous and smooth morphology in perovskite films having an increased optical path length. Carrier extraction is still limited in thick films made using the method; but we show that the addition of a Lewis base decreases the trap density and enhances the electron diffusion length so that it exceeds 2 µm. As a result, we report a semi-transparent perovskite top cell that provides a stabilized PCE of 19% and an average NIR transmittance (800–1100 nm) of 85%. As bottom cells, CQD:organic hybrid cells and silicon heterojunction (SHJ) cells are tailored to enhance their NIR photocurrent generation. With these top and bottom cells, we achieve a PCE of 24% in perovskite/CQD tandems and 28.2% in perovskite/silicon tandems.

## Results

### Estimating optical loss in semi-transparent perovskite cells.
To quantify the additional optical losses originating from using transparent conductive electrodes (TCEs) as back electrodes, we employed transfer-matrix methods (TMM) simulations in rear-mirror and semi-transparent devices. We calculated the response for PSCs made of three most commonly used perovskite compositions. These devices show 8–10% current loss for semi-transparent devices in the typical thickness range of 300–500 nm[29] as implemented in PSCs (Supplementary Fig. 1), which can be reduced to below 2% when the semi-transparent devices are around 300 nm thicker than their opaque counterparts. We conclude that thick films with sufficiently long carrier diffusion length may enable high-efficiency semi-transparent cells.

### Forming thick perovskite films with smooth morphology.
We focused on the triple-cation $Cs_{0.05}FA_{0.81}MA_{0.14}PbI_{2.55}Br_{0.45}$ (CsFAMA) perovskite because of its high-performance in opaque rear-mirrored devices and lower chances to form a rough surface[19,25,30–32]. For spin-coated perovskite films, there are two ways to increase thickness[33]. When we increased perovskite precursor solution concentration, the transition from intermediate states to the perovskite phase is fast because of the high supersaturation of precursor solution and rapid perovskite reaction rate during subsequent thermal annealing[34,35]. However, such a quick intermediate to perovskite phase transition induces compressive stress because of sudden volume change during the process, resulting in films with a rough and wrinkled surface[24,25], as shown in Fig. 1a–c. Despite the increased average thickness of the films from 400 to 700 nm, the high surface roughness and a significant amount of $PbI_2$ precipitation negatively affect the device performance (Fig. 1e–g). Another option is decreasing the spin-coating speed. The combination of 1.4 M precursor with reduced spinning rotation per minute (RPM) results in around 700 nm thickness perovskite film with a smoother surface (Fig. 1d, f). However, the performance improvement is modest and PCE still trails that of 400 nm devices (Fig. 1g).

Scanning electron microscope (SEM) images reveal pinholes in perovskite films prepared by low spin-coating speed (control), a finding previously reported to arise due to insufficient nucleation (Fig. 2a)[34]. For the one-step perovskite fabrication method, the crystallization kinetics are largely dependent on the anti-solvent dripping process. We reasoned that the slower rotation speed makes it challenging to yield compact films. The narrow anti-solvent dripping time windows (for a burst of intermediate phase nucleation)[17,34] require fast spreading and quick evaporation of anti-solvent across the surface of the spinning film, which is not achievable at low RPM. The reduced centrifugal force decreases the lateral flow of the anti-solvent and hence the ability to spin-off excess polar solvents. As a result, the transition from perovskite precursor solution to solid-state intermediate phase is incomplete. Due to the inhomogeneous supersaturation across the whole spinning substrate, only a small portion of the converted film is ready for perovskite-phase formation. Indeed, sparse perovskite phases are observed across the substrate, which translates to a less compact perovskite film with pinholes after thermal annealing (Fig. 2d).

This led us to the view that the fast removal of excess solvent coupled with low-speed spinning could potentially produce thicker films with a homogenous morphology. In this approach, we raised the spin-coating speed to higher RPM precisely 1 s before anti-solvent dripping. This was intended to increase the lateral hydrodynamic flow and evaporation rate of the mixed solvents to facilitate the rapid nucleation of the intermediate phase. The goal of the technique—which we term boosted solvent extraction (BSE)—was to form homogeneous and smooth films at reduced spin speed (Supplementary Fig. 2). The films prepared by the BSE technique show a more compact morphology (Fig. 2b, e) with lower surface roughness (Supplementary Fig. 3). X-ray diffraction patterns of intermediate phases in the unannealed control film show broader peak widths, which also indicate the existence of excess solvent, as the local structural order of the intermediate phase can be disturbed by DMSO/DMF over-incorporation (Fig. 2c). The improved surface morphology of BSE films enables a better photovoltaic performance and paves the way for subsequent studies (Fig. 2e).

We then prepared perovskite films with thicknesses ranging from 400 to 700 nm (Fig. 3a–c); the corresponding devices demonstrate the anticipated $J_{sc}$ improvements, as evidenced by the EQE spectra in Fig. 3f. The EQE integrated $J_{sc}$ values are 19.3, 20.1, 21, 21.8 mA cm$^{-2}$ for semi-transparent devices with

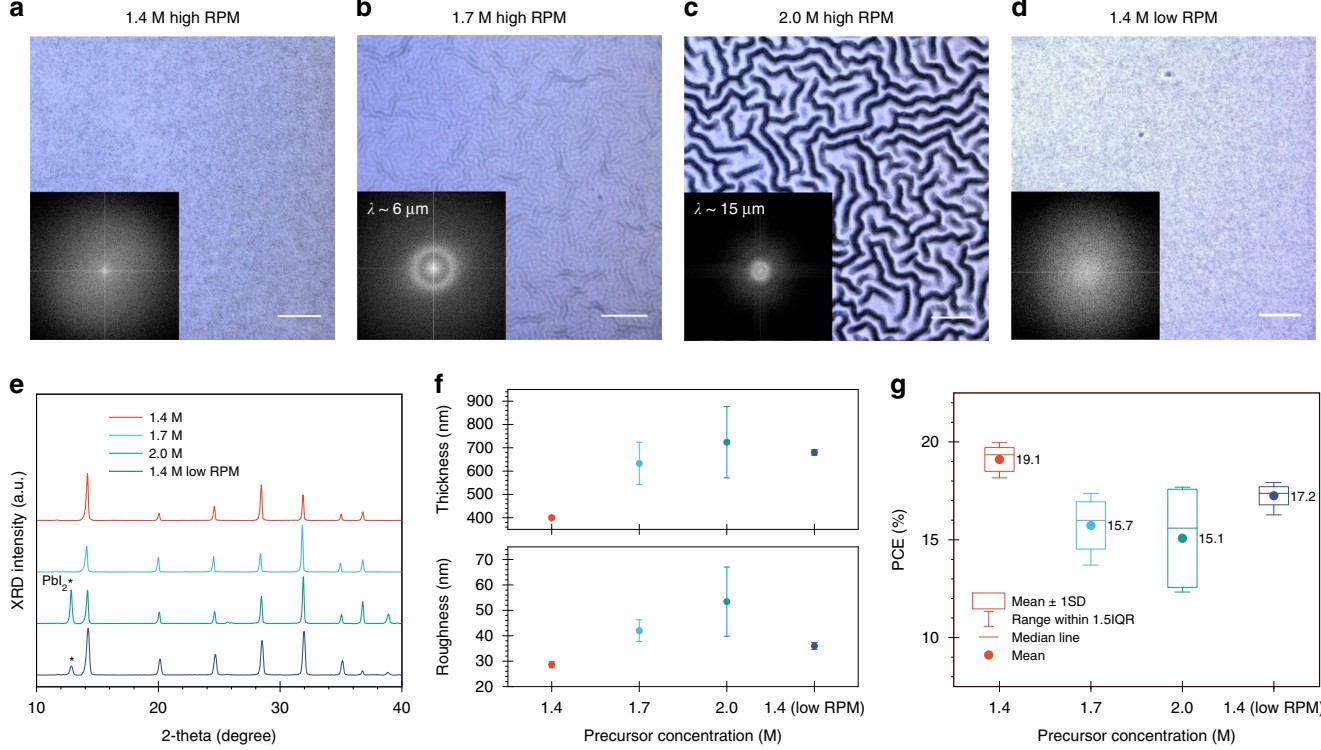

**Fig. 1 Morphology and photovoltaic performance of perovskite films prepared by different precursor concentrations. a–d** Optical microscope images and corresponding fast Fourier transform (FFT) of perovskite films prepared by different methods ($\lambda$ is the wrinkling wavelength). Scale bar: 10 μm. **e** XRD patterns and **f** surface morphology of perovskite films. Error bars are standard deviations of ten locations in SEM cross-sectional images for thickness and of root mean squared (RMS) analysis from AFM images for roughness. **g** Corresponding PCE values (12 devices each).

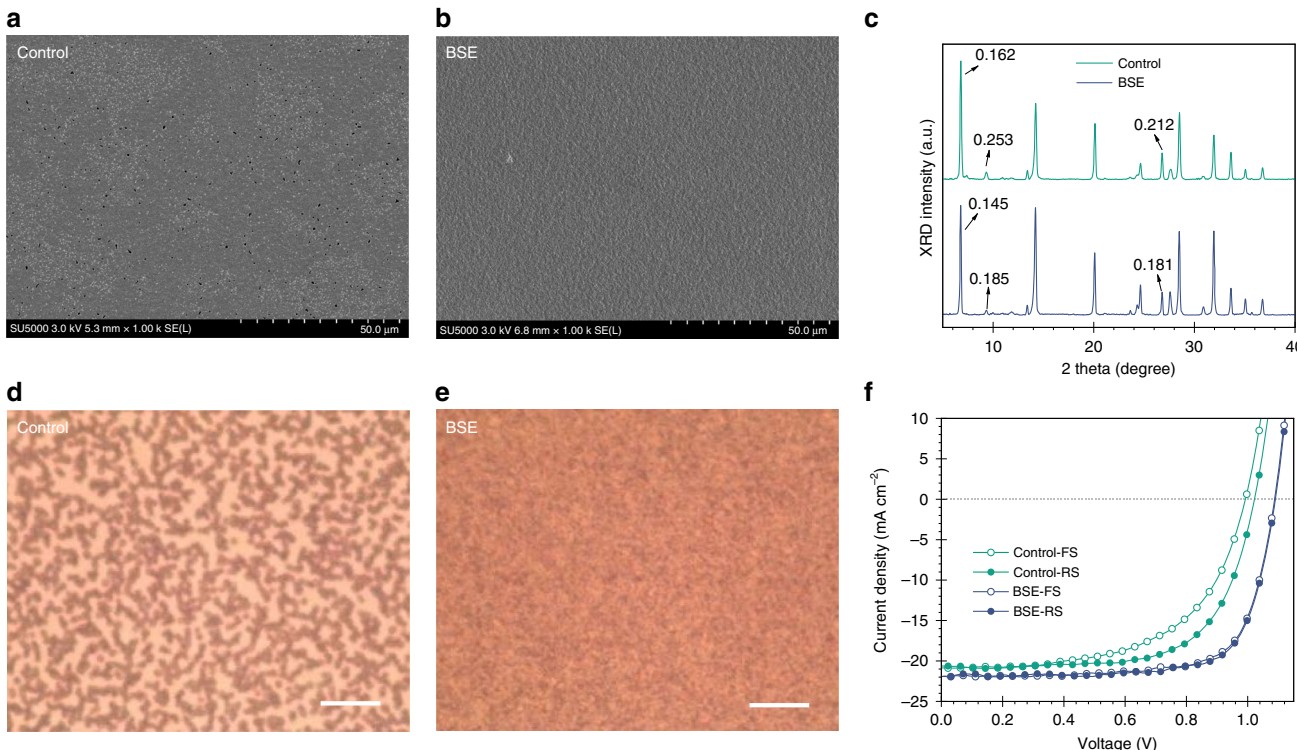

**Fig. 2 Morphology and photovoltaic performance of perovskite films prepared by boosted solvent extraction (BSE) and non-BSE methods. a, b** SEM images of perovskite films prepared using non-BSE and BSE methods; all images are acquired following thermal annealing. **d, e** Optical microscope images showing perovskite films prepared by non-BSE and BSE methods before thermal annealing. Scale bar: 10 μm. **c** X-ray diffraction pattern of perovskite films prepared by non-BSE and BSE methods before thermal annealing. Numbers are full width at half maximum. **f** JV curves of devices based on the BSE and non-BSE perovskite films.

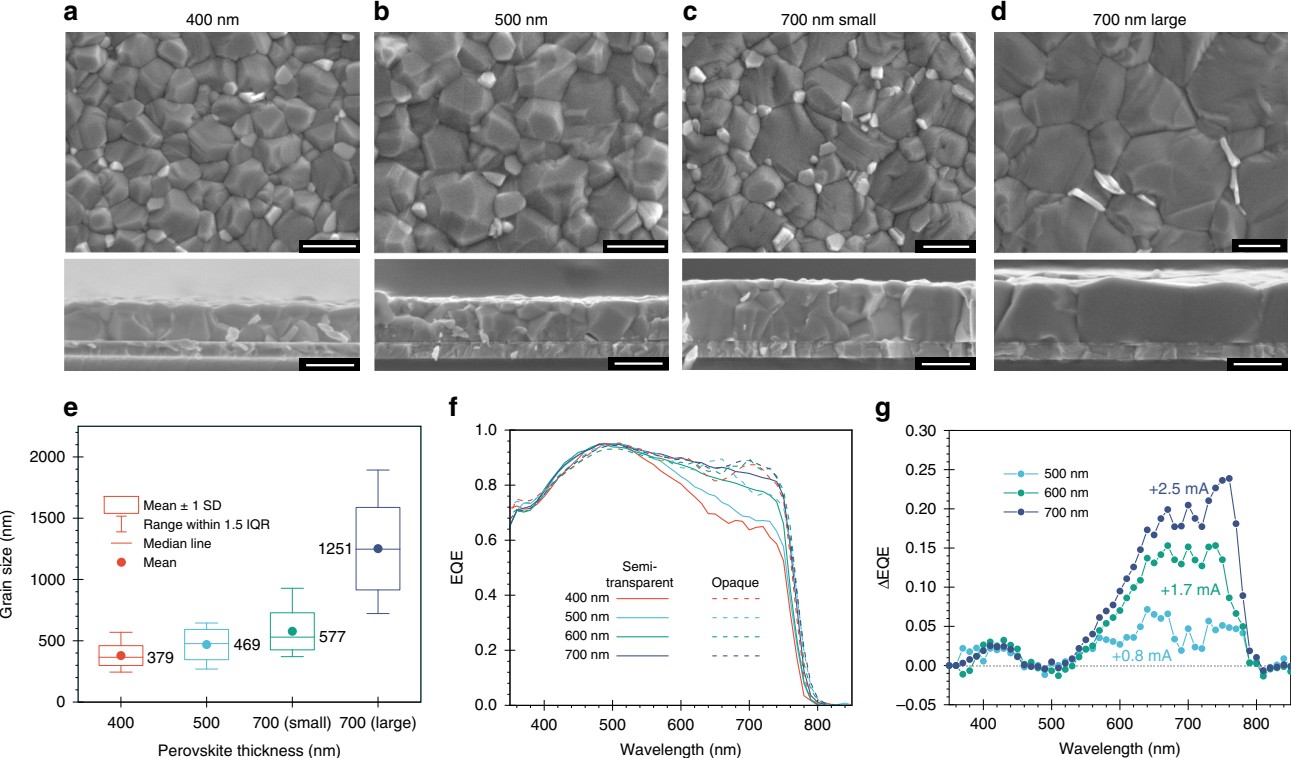

**Fig. 3 Effects of perovskite thickness on photon collection. a–d** Top and cross-sectional SEM views of perovskite films with different thicknesses (**a** 400 nm, **b** 500 nm, **c** 700 nm, **d** 700 nm with larger grains) deposited onto glass/ITO/SnO₂ substrates. **e** Statistical distribution of grain size obtained by SEM images from **a–d** films. The scale bar is 500 nm. **f** Measured EQE spectra of semi-transparent (solid line) and back-mirrored (dashed line) devices with varying perovskite absorber layer thickness. **g** Difference of EQE spectra between thicker perovskite devices compared to control device with 400 nm thickness.

absorber thickness of 400, 500, 600, and 700 nm, respectively. Three phenomena are observed: (1) the increased $J_{sc}$ values of thicker devices are a direct result of the enhanced EQE in the wavelength range from 500 to 780 nm, as compared to a 400 nm thick control device. (2) Semi-transparent devices do not show interference features due to the minimized back reflection, which is evident in opaque rear-mirror devices. (3) Interference only happens between the 550 and 750 nm wavelength range, due to the reduced absorption coefficient of the perovskite layer at the band edge, compared to the Beer-Lambert regime for short wavelengths (below 500 nm)[36]. The EQE difference of devices with respect to the control cell shows that 500, 600, and 700 nm thick devices gain an integrated $J_{sc}$ of 0.8, 1.7, and 2.5 mA cm⁻², respectively (Fig. 3g). The improved $J_{sc}$ results confirm that the thickness of the perovskite absorber is critical to enhancing light harvesting in the semi-transparent devices by compensating for the loss from using TCEs as rear electrodes.

**Reducing traps in thick perovskites films.** Short carrier diffusion lengths have been a limiting factor for thick films perovskite devices[26–28]. We found that the reduced carrier lifetime in 700 nm films (Fig. 4a) was correlated to the multi-grain structure between the charge transport layers (Fig. 3c). This led us to believe that a lower density of grain boundaries could help improve carrier transport, since they are regarded as trapping sites for carrier recombination[37–39]. We employed urea, a Lewis base additive, in an attempt to increase the grain size[35]. Figure 3e shows the statistical distribution of the apparent lateral grain size from SEM of perovskite films fabricated under different conditions[39]. Urea-treated 700 nm thick perovskite films exhibit a larger apparent grain size (700 nm-large), averaging 1.3 µm

compared to 0.6 µm without any additive (700 nm-small). The larger grain enables a single perovskite domain structure through the vertical direction of the active layer (Fig. 3d). We obtained the 1D carrier diffusion length ($L_D$) with the aid of time-resolved photoluminescence spectroscopy (TRPL)[40,41]. The 700 nm-small films show greatly reduced carrier collection efficiency compared to the 400 nm reference film, evident from the longer carrier lifetime (from 46 to 162 ns) in spiro-OMeTAD-quenched TRPL measurement (Fig. 4b). However, the 700 nm-large film recovers the fast quenching observed in the thinner sample, by reducing the lifetime to 46 ns. In the ETL-quenching experiment using phenyl-C61-butyric acid methyl ester (PCBM), the 700 nm-large film shows the fastest PL decay at 180 ns, suggesting the best electron extraction efficiency among the three films (Fig. 4c). By comparing the quenched and unquenched TRPL lifetimes, we found that $L_D$ of the hole is estimated to be over 1000 nm for all films and not a limiting factor for carrier transport. On the contrary, $L_D$ of the electron in 700 nm-small films is calculated to be only $571 \pm 10$ nm, which is shorter than its thickness and considerably less than that of 700 nm-large films ($2347 \pm 46$ nm).

The 700 nm-large devices show a higher average reverse scan PCE of 18.4%, compared to 17.3% for the same thickness but smaller grain size (Supplementary Fig. 4a). While the $J_{sc}$ does not change, the advantages of enhanced grain size are readily seen in the improved open-circuit voltage ($V_{oc}$) and fill factor (FF). The longer carrier diffusion length in the 700 nm-large film leads to an improved FF of 75 from 72%. Fast transient photocurrent (TPC) decay confirms the improvement of charge collection in the treated device (Fig. 4d). Fewer grain boundaries in the vertical direction reduce carrier recombination, as evidenced by the longer transient photovoltage (TPV) lifetime (Fig. 4e) and smaller trap-filled limit voltage ($V_{TFL}$) from space charge limited current

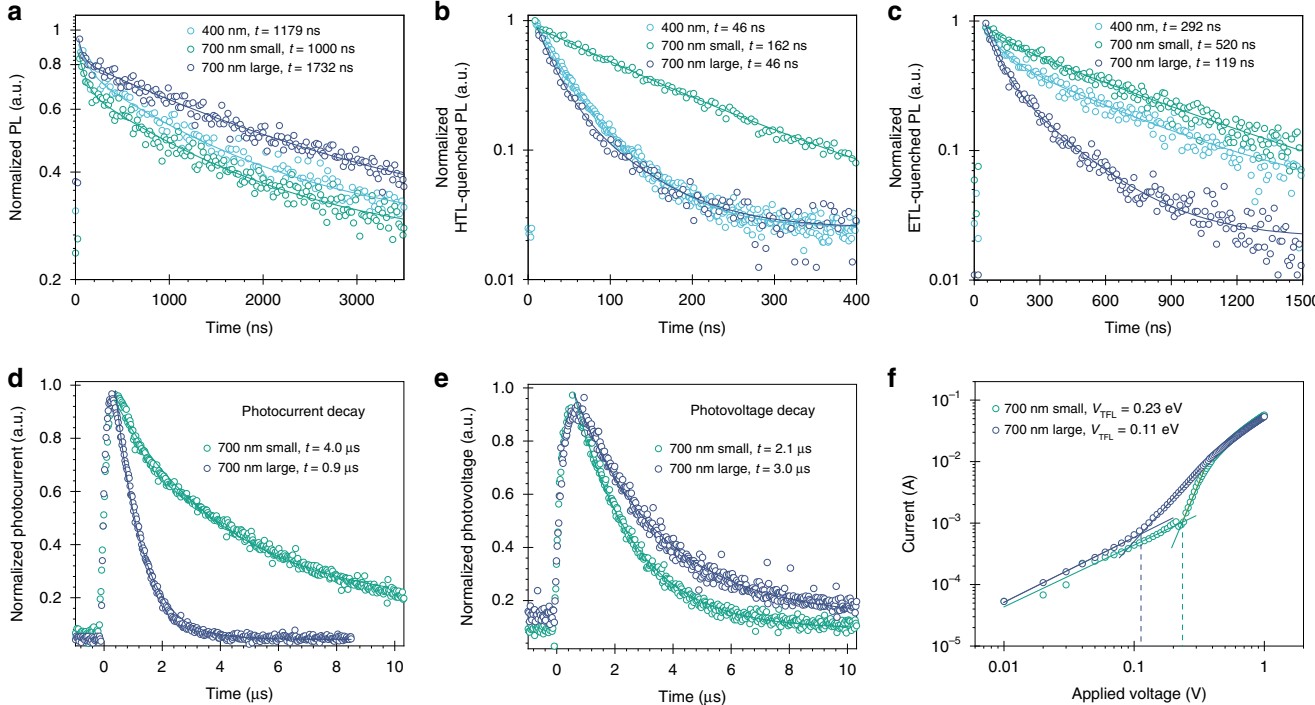

**Fig. 4 Effects of grain size on carrier transport and trap density in thick perovskite. a** Non-quenched, **b** spiro-OMeTAD, and **c** PCBM-quenched time-resolved photoluminescence spectroscopy measurements on perovskite films. **d** Transient photocurrent, **e** photovoltage decay and **f** Space charge limited current of 700 nm perovskite devices with various grain sizes.

(SCLC) measurement (Fig. 4f), in electron-only devices (ITO/SnO₂/Perovskite/PCBM/Ag), resulting in a $V_{oc}$ increase of 20 meV. A benefit related to TCEs compared to metal electrodes is improved device operation stability[42–44]. In our semi-transparent cells, TCEs are implemented as working electrodes and also act as effective sealants, protecting our devices. Both large-grain and small-grain devices show good operational stability, retaining performance to within 75% of initial (post-burn-in) performance following more than 1000 h under AM1.5 G maximum power point (MPP) tracking in N₂ atmosphere period (Supplementary Fig. 4b). Large-grain devices show slightly better stability than the small grain counterpart, most likely due to the reduced grain boundaries, which are vulnerable to moisture ingress[34]. The champion device exhibits a 19% stabilized power output (Supplementary Fig. 4c). To the best of our knowledge, this is by far the most efficient semi-transparent solar cells reported, with 85% average transmittance in the NIR region (Supplementary Fig. 4d). One of the key enablers of high NIR transmittance is replacing commercial ITO with previously-developed highly-conductive Zr-doped In₂O₃ (IZrO) TCOs, whose parasitic free carrier absorption is suppressed, for a given free carrier density, by virtue of its enhanced carrier mobility (Supplementary Fig. 5)[45].

**Designing efficient bottom cell structures**. Bottom cells that have strong and spectrally broad absorption beyond the perovskite band-dedge, are crucial to the realization of highly efficient tandem solar cells. We used a solution-processed bottom cell: a colloidal quantum dot (CQD):organic hybrid structure. We took advantage of the bandgap tunability of PbS CQDs across visible to infrared wavelengths (0.7–1.3 eV) via size-control[46] and their high EQEs (80% near 1100 nm)[47], which makes them suitable for bottom cells in a tandem configuration. In addition, we stacked NIR-sensitizing organic molecules to complement the intrinsic absorption deficiency of CQDs near the excitonic valley. For that purpose, emerging non-

fullerene acceptors (NFA), IEICO-4F (2,2′-((2Z,2′Z)-(((4,4,9,9-tetra-kis(4-hexylphenyl)-4,9-dihydro-s-indaceno[1,2-b:5,6-b′]dithiophene-2,7-diyl)bis(4-((2-ethylhexyl)oxy)thiophene-5,2-diyl))bis(methanylylidene))bis(5,6-difluoro-3-oxo-2,3-dihydro-1H-indene-2,1-diylidene))dimalononitrile), is combined with the CQDs, forming the CQD/organic hybrid structure[48]. Absorbance spectra demonstrate that IEICO-4F exhibits strong absorption across 600–1000 nm, which complements the CQD absorption (Fig. 5a).

The best hybrid devices surpass CQD-only solar cells (12.1%) and showed a PCE of 13.0%, the highest solar cell efficiency among CQDs and CQD:organic hybrids[49]. $J_{sc}$ was improved from 28.6 to 31.3 mA cm⁻², a relative enhancement of 9.5% which is in good agreement with estimates from simulations (Supplementary Fig. 6). We acquired spectral responses to confirm the origin of $J_{sc}$ improvement (Fig. 5c). The hybrid devices showed improved EQE from 700 to 900 nm, where the NFA absorbs (Supplementary Fig. 7). Overall, the EQE is over 80% up to 1150 nm, indicating that CQD:organic hybrid structures are advantageous as bottom cells for tandem applications.

We applied a similar design strategy of tailored NIR photon collection to SHJ cells. In the 4T configuration, the silicon bottom cell works at low current density due to the filtering by the perovskite top cell. To maximize the current output, we designed a specific contact geometry that minimizes the shadowing losses at a low light intensity, preserving at the same time optimal contact properties. In doing this, we decreased the number of metal fingers to reach an overall 2.5% metal coverage ratio. To improve the NIR response of the devices, we replaced the ITO electrode with H-doped In₂O₃ (In₂O₃:H) to reduce free-carrier absorption (Fig. 5d), at the cost of increased reflection in the blue (Fig. 5e). Fortunately, this reflection does not result in a performance loss on the overall 4T tandem solar cells since the majority of the blue part of the spectrum is already absorbed by the perovskite top cells. By taking into account all of these improvements, the spectral response of the SHJ cell maximizes the NIR absorption (700–1200 nm) at the expense of decreased

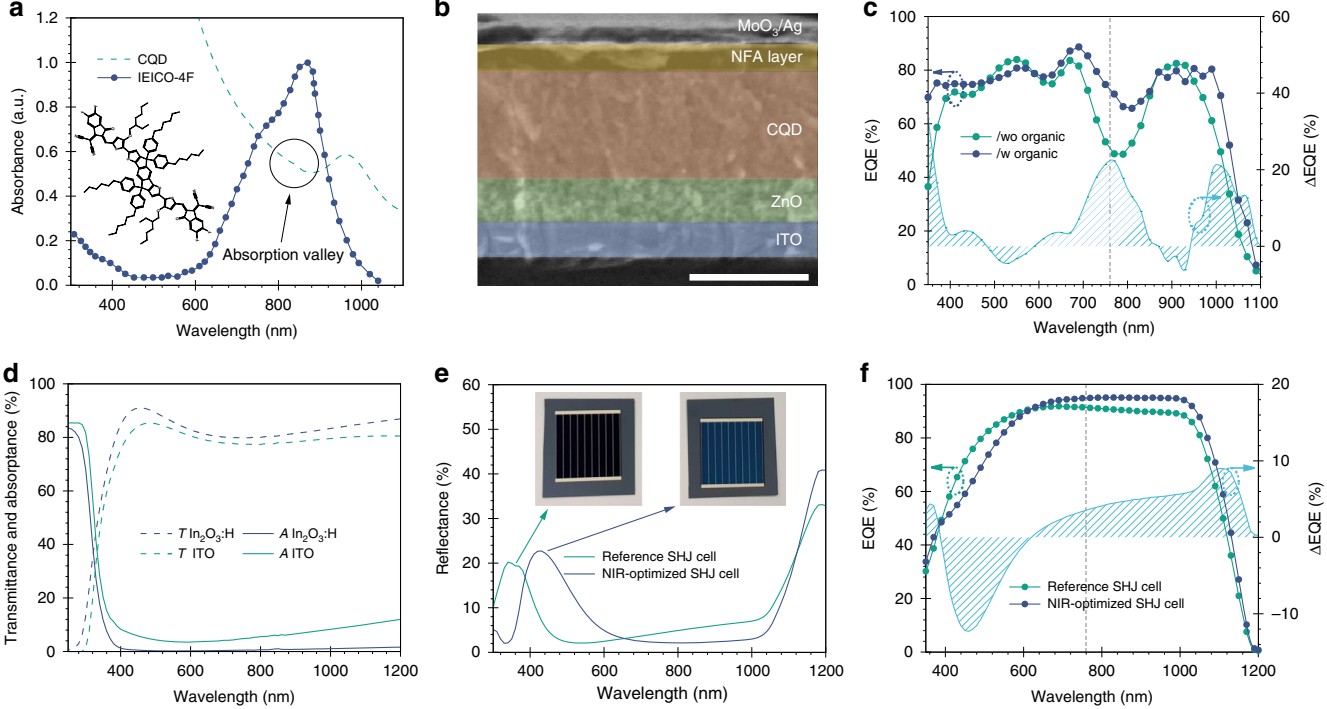

**Fig. 5 NIR-tailored design for bottom cells. a** Molecular structure and absorbance spectrum of NFA. The symbol line denotes the absorbance of CQD. **b** Cross-sectional SEM image of fully solution-processed CQD bottom cell. Scale bar: 500 nm. **c** EQE spectra of CQD device without and with the organic layer. **d** Transmission (dashed) and absorption (solid) of $In_2O_3$:H and standard ITO films. **e** Reflection and **f** EQE spectra of reference and NIR-optimized SHJ cell. Short dashed lines indicate the bandedge cut-off of perovskite front cell.

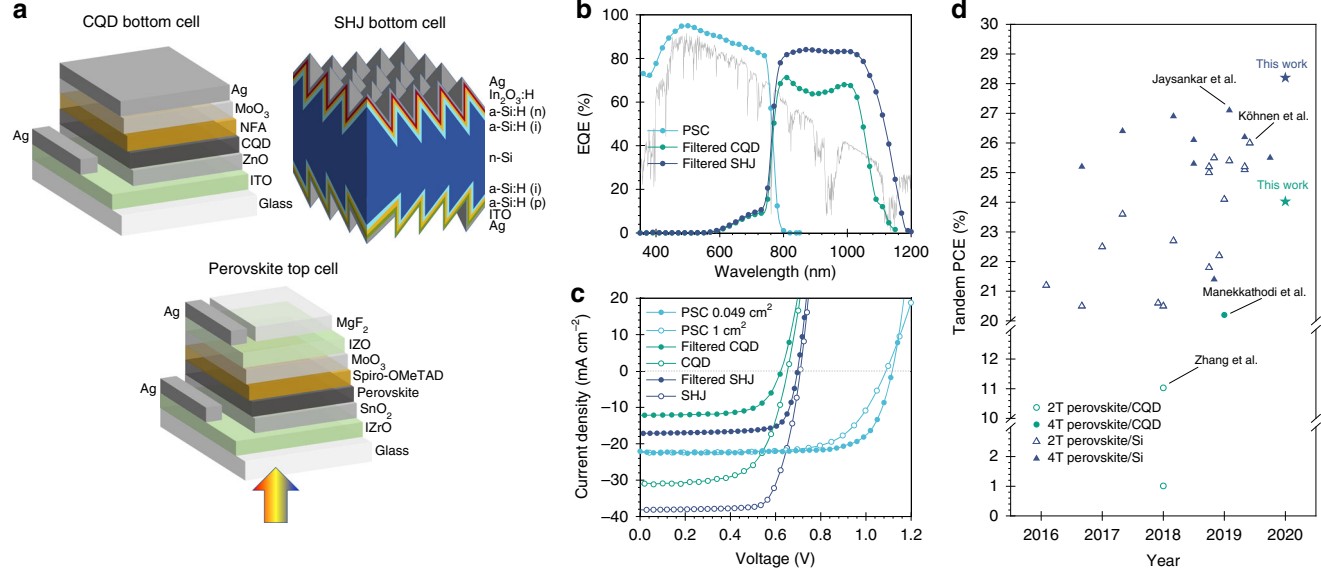

**Fig. 6 Design and photovoltaic performance of mechanically stacked tandem solar cells. a** Device structures of perovskite top cell, CQD, and SHJ bottom cell. **b** EQE spectra (overlaid with AM1.5 G) and **c** JV curves of perovskite top cell, CQD:organic hybrid bottom cell and SHJ bottom cell in the 4T tandem configuration. **d** PCE progress of perovskite-based tandem solar cells.

efficiency in the visible (400–600 nm) range, leading to an improved $J_{sc}$ from the SHJ as a bottom cell compared to the reference SHJ designed to work at one sun condition (Fig. 5f).

Encouraged by CQD:organic hybrid and silicon bottom cells with improved NIR responses, we stacked these with BSE and grain-engineered semi-transparent top cells to form 4T tandem cells (Fig. 6a). Figure 6b, c combine the 4T EQE spectrum and JV curves of each constituent cell in the 4T structure. The PV

parameters are summarized in Table 1. The perovskite/CQD system demonstrates highly efficient photon-harvesting capability as an all-solution processed tandem, yielding a $J_{sc}$ of 34.5 mA cm² and a PCE of 24%. The perovskite/silicon pair synergistically generates a current density of 39.5 mA cm² in tandem, leading to a PCE of 28.2%, which is the highest in the perovskite-silicon tandem category (Fig. 6d and Supplementary Table 1). We also fabricated 1 cm² size tandem, which yields a PCE of 25.7%. This

**Table 1 Photovoltaic parameters of PSC top cell, CQD, and SHJ bottom cells.**

| Cells | $V_{oc}$ (V) | $J_{sc}$ (mA cm$^{-2}$) | FF (%) | PCE (%) | Stabilized (%) |
|---|---|---|---|---|---|
| *Semi-transparent PSC* | | | | | |
| (0.049 cm$^2$) | 1.12 | 22.3 | 77.7 | 19.4 | 19.0 |
| (1 cm$^2$) | 1.10 | 22.4 | 66.9 | 16.5 | 16.5 |
| Filtered CQD | 0.62 | 12.2 | 66 | 5.0 | 5.0 |
| *PSC-CQD 4T* | | | | | |
| (0.049 cm$^2$) | | | | 24.4 | 24.0 |
| Filtered SHJ | 0.70 | 17.2 | 76.6 | 9.2 | 9.2 |
| *PSC-SHJ 4T* | | | | | |
| (0.049 cm$^2$) | | | | 28.6 | 28.2 |
| (1 cm$^2$) | | | | 25.7 | 25.7 |

is limited by fill factor due to increased series resistance from TCEs in the semi-transparent PSC (Supplementary Fig. 8), and it will in future become important to incorporate metal fingers as busbars.

## Discussion

The transparency requirement of semi-transparent solar cells, taken together with the lower extinction coefficient near the absorption bandedge of perovskite film, calls for different design rules from those employed in conventional single-junction devices. The boosted solvent extraction technique allows us to extend the optical path length in semi-transparent perovskite cells. By then enhancing the electron diffusion length to overcome decreased carrier transport in thicker films, we achieved 19% PCE semi-transparent top cells with minimized current loss compared to opaque rear-mirrored devices. The improved NIR response in bottom cells allowed us to demonstrate superior photon collection in all-solution-processed tandems with perovskite/CQD devices (34.5 mA cm$^2$), and the highest PCE among literature-reported perovskite/silicon tandem cells (28.2%).

## Methods

**Materials**. Commercial ITO substrates (20 Ω sq$^{-1}$) with 25 mm × 25 mm dimension were purchased from TFD Inc. PbI$_2$ beads, PbBr$_2$, CsI, and MoO$_3$ powders were purchased from Alfa Aesar. Methylammonium iodide (MAI), formamidinium iodide (FAI), methylammonium bromide (MABr) were purchased from GreatCell Solar. The SnO$_2$ nanoparticle was bought from Alfa Asesar. Spiro-OMeTAD was purchased from Xi'an PLT. Lithium bis(trifluoromethane)sulfonimide (LiTFSI), 4-tert-Butylpyridine (tBP) were from Sigma-Aldrich. All the solvent used in the process were anhydrous solvents from Sigma-Aldrich. IZO and IZrO sputtering targets was purchased from Kurt J. Lesker and Plasmamaterials, respectively.

**Electron-transport layer (ETL) fabrication**. Patterned ITO substrates were cleaned for 30 mins in ultrasonication with deionized water, acetone, and iso-propanol, respectively. The precursor for SnO$_2$ layer fabrication was prepared by diluting (1:6 ratio) the SnO$_2$ nanoparticles solution (Alfa Asear) in deionized water. This solution is spin-coated on the ITO substrates with 3000 RPM for 15 s and annealed at 150 °C for 30 min. Before spin-coating, ITO substrates were treated in O$_2$ plasma for 10 min.

**Perovskite absorber layer fabrication**. The perovskite absorber layers were deposited inside the N$_2$-filled glove box with a controlled H$_2$O and oxygen level to less than 1 PPM. The temperature inside was monitored to be 25–28 °C. The precursors for Cs$_{0.05}$FA$_{0.81}$MA$_{0.14}$PbI$_{2.55}$Br$_{0.45}$ perovskites were prepared by dissolving the PbI$_2$, PbBr$_2$, CsI, FAI, and MABr in DMF:DMSO (4:1 in volume) solvents, as previously reported[19]. For Lewis-base treated perovskite films, 5–10 mg mL$^{-1}$ urea was added in the perovskite precursor solutions. For high RPM devices, the substrate was spun at 1000 RPM for 10 s with an acceleration of 200 RPM at first, and then at 6000 RPM for the 50 s with an acceleration of 2000 RPM. In the second step, 50 µL chlorobenzene was dropped onto the substrate during the last 5 s of the spinning. For low RPM control devices, perovskite films were deposited on to the SnO$_2$ coated ITO substrates following a two-step spin-coating procedure. The substrate was spun at 200 RPM for 2 s with an acceleration of 300 RPM at first, and then at 2000 RPM for the 60 s with an acceleration of 2000 RPM. In the second step, 150 µL chlorobenzene was dropped onto the substrate during the last 5 s of the spinning. For low RPM BSE devices, perovskite films were

deposited on to the SnO$_2$ coated ITO substrates following a three-step spin-coating procedure. The substrate was spun at 200 RPM for 2 s with an acceleration of 300 RPM at first, and then at 2000 RPM for the 55 s with an acceleration of 2000 RPM. For the third step, RPM is set to 6000 with the acceleration of 10,000 RPM s$^{-1}$ for 5 s, and 150 µL chlorobenzene was dropped onto the substrate immediately after RPM reached 6000. The substrate was immediately placed on a hotplate and annealed at 100 °C for 25 min.

**Hole-transport layer (HTL) fabrication**. The HTM (Spiro-OMeTAD) was deposited immediately after the fabrication of perovskite films. Spiro-OMeTAD solution was prepared by dissolving 67.5 mg Spiro-OMeTAD in 1 mL chlorobenzene. Then 35 µL LiTFSI solution (350 mg mL$^{-1}$ in acetonitrile) and 25 µL tBP were added. This solution was spun at 4000 RPM for 30 s on perovskite films. Then the samples were kept in dry box with desiccant overnight at dark for oxidation.

**Back electrode fabrication**. Ten nanometer of MoO$_3$ was deposited atop the Spiro-OMeTAD by thermal evaporation followed by 120 nm thermally evaporated Ag as a back contact electrode for opaque perovskite solar cell. The device area (0.053 cm$^2$) is defined by a deposition mask. 100 nm IZO electrode was sputtered in Angstrom Next dept system with a processing pressure of 1.5 mTorr, power density is 9 W cm$^{-2}$. The gas flow in the sputtering chamber was Ar mixed with 0.4% O$_2$ at 15 SCCM. The resultant IZO films have sheet resistance of 30–50 Ω sq$^{-1}$. Ten nanometer of MoO$_3$ was thermally evaporated at a rate of 0.2 Å s$^{-1}$ onto Spiro-OMeTAD layer as a buffer layer before IZO sputtering.

**CQD:organic hybrids bottom cell fabrication**. Oleic acid-capped PbS CQDs (1st excitonic peak at 950 nm) were prepared as previously described[9]. For the fabrication of hybrid devices, a ZnO nanoparticle solution was spun onto a pre-cleaned ITO substrate at 3000 RPM for 20 s twice, forming a 150 nm-thick layer. For CQD layer, CQDs were exchanged by the PbX$_2$/NH$_4$Ac solution-phase ligand-exchange process. (PbI$_2$: 0.1 M and PbBr$_2$: 0.02 M and NH$_4$Ac: 0.04 M). Exchanged dots are precipitated and dried in the vacuum chamber for 10 min, followed by re-dispersion in butylamine (BTA) at a concentration of 330 mg mL$^{-1}$. Prepared CQD inks spun onto ZnO layer at 2500 RPM for 30 s. After one-day air drying, an organic solution (PTB7-Th:IEICO-4F = 1:4, 12 mg mL$^{-1}$ in chlorobenzene:1-chloronaphthalene = 96:4 wt%) was deposited at 2000 RPM for 60 s followed by drying process at 80 °C for 5 min. Finally, the MoO$_3$ (10 nm)/Ag (120 nm) electrode was deposited using thermal evaporation under a high vacuum (below 10$^{-6}$ Torr) through a 0.053 cm$^2$ area shadow mask.

**Fabrication of the silicon bottom cell**. SHJ bottom cells were fabricated on float-zone double-side-textured four inches wafers (n-doped, resistivity 1 to 5 Ω cm$^{-1}$, thickness 250–280 µm). The texturing process was done in an alkaline solution to get randomly distributed pyramids. Subsequently, the wafers were cleaned in RCA1 and RCA2 solution. Prior to the PECVD depositions, the wafers were dipped in 5% hydrofluoric acid solution to remove the native oxide layer. Then, the wafers were passivated on both sides with an intrinsic layer of amorphous silicon (i) followed by the doped silicon amorphous layers, n on the front and p on the back. The i, p, and n layers were deposited in a PECVD cluster (Indeotec Octopus 2) with a thickness of 8, 7, 15 nm respectively. The back contact of the SHJ was realized by sputtering sequentially ITO and Ag (150 and 250 nm, respectively) in the PVD part of the Octopus 2 cluster. After the In$_2$O$_3$:H top contact deposition, metal fingers are screen printed using a commercial silver paste and immediately annealed at 200 °C for 15 min in ambient air.

**Deposition of the IZRO films**. IZRO films (100 nm) were deposited by radio-frequency sputtering technique on pre-cleaned quartz substrates using Angstrom Engineering EvoVac sputtering tool. The deposited films were annealed in ambient air at 200 °C for 25 min. The resultant films had sheet resistivity of 25 Ω sq$^{-1}$. The details of the deposition parameters have been reported in the previous study[45].

**Device characterization**. The JV curves under a simulated AM1.5 solar spectrum were acquired in an N$_2$ purging atmosphere with 0.049 cm$^2$ aperture mask. Large area devices have active area of 1.95 cm$^2$, and 0.49 cm$^2$, 1 cm$^2$ and 1.68 cm$^2$ masks were used for accurate JV measurements. The scanning rate is 100 mV s$^{-1}$. The input power density was adjusted to 1 sun using a NIST-traceable calibrated reference cell (Newport 91 150 V). The lamp spectrum was measured using irradiance-calibrated spectrometers. For EQE calibration, two photodetectors, Newport 818-UV and Newport 838-IR were used. The current response was collected at short circuit conditions with a Lakeshore preamplifier connected to a Stanford Research 830 lock-in amplifier. The stabilized power output was obtained by setting the bias voltage to the initial $V_{MPP}$ that was determined from the JV curve.

During the perovskite/CQD 4T measurement, the CQD device was masked by 0.049 cm$^2$ and placed behind the actual working perovskite device with an active area of 0.053 cm$^2$. JV measurements of the 4T perovskite/silicon tandem solar cells have been performed using Wavelabs Sinus 220 LED-based solar simulator with AM 1.5 G irradiance spectrum at a controlled temperature of 25 °C. The silicon cell

device area was covered by a larger (2 cm × 2 cm) semitransparent perovskite cell filters, fabricated in the same batch and procedure as that of the actual devices. The illumination area of the devices was determined by a laser-cut aperture with 2.32 cm$^2$ area. EQE measurements have been performed using PV-Tools LOANA without any light and voltage biasing. The beam size was 15 by 15 mm during the EQE measurements, which means the measured data includes SHJ metal fingers as well. Perovskite/CQD 4T characterizations have been performed in $N_2$ purging environment while perovskite/silicon 4T were measured in ambient air with around 50% RH and without any encapsulation.

**Materials characterization**. The absorbance was characterized via UV–Vis–IR spectrophotometer (Lambda 950). The cross-section images of devices were obtained by Hitachi SU5000 microscope. Photoelectron spectroscopy was carried out in a PHI5500 Multi-Technique system with non-monochromatized He-Iα radiation (UPS) (hν = 21.22 eV). All samples were prepared on ITO substrate.

Steady-state PL and time-resolved PL were measured using a Horiba Fluorolog time-correlated single-photon-counting system with photomultiplier tube detectors. The light was illuminated from the top surface of the perovskite film. For steady-state PL measurements, the excitation source is a monochromated Xe lamp (peak wavelength at 540 nm with a line width of 2 nm). For time-resolved PL, we used a green laser diode ($\lambda = 540$ nm) for the excitation source with an excitation power density of 5 mW cm$^{-2}$. The PL decay curves were fitted with biexponential components to obtain a fast and slow decay lifetime. The mean carrier lifetimes τ for the biexponential fit was calculated by the weighted average method.

Transient photovoltage decays were measured on a home-made system. For the transient photovoltage decay measurements, a red-light emitting diode was used to modulate the $V_{oc}$ with a constant light bias. The pulse duration is set to 1 μs and the repetition rate to 500 Hz. For the constant light bias, a continuous light source from a Xe lamp was coupled through a fiber to collimate on the active area of the solar cell under study. The intensity of the pulsed laser was set in a way that the modulated $V_{oc}$ was 50 mV to ensure a perturbation regime. The open-circuit voltage transient, induced by the light perturbation was measured with a digital oscilloscope set to an input impedance of 1 MΩ. The charge recombination lifetime was fitted by a single exponential decay.

**Optical modeling**. All refractive indexes were obtained by variable angle spectroscopic ellipsometry and homemade MATLAB code was used for building up transfer matrix formalism. We assumed that each layer was optically flat and have no other scattering effect. Spectroscopic ellipsometry was performed using a Horiba UVISEL Plus extended range ellipsometer with a 200 ms integration time, a 5 nm step size and a 1 mm diameter spot size at an incident angle of 50, 70, and 90 degrees. Soda-lime glass slides were used as substrates for each material, with their back covered with cloudy adhesive tape to ensure back-reflections are diffusively reflected away from the detector. Fitting was performed using Horiba's DeltaPsi2 software.

**Reporting summary**. Further information on experimental design is available in the Nature Research Reporting Summary linked to this paper.

## Data availability
The data that support the findings of this study are available on reasonable request from the corresponding author.

## Code availability
All custom codes used the current study are available from the corresponding author on reasonable request.

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

## Acknowledgements

This research was made possible by Ontario Research Fund-Research Excellence program (ORF7-Ministry of Research and Innovation, Ontario Research Fund-Research Excellence Round 7); and by the Natural Sciences and Engineering Research Council (NSERC) of Canada. This work was also supported by the King Abdullah University of Science and Technology (KAUST) Office of Sponsored Research (OSR) under Award No. OSR-2018-CPF-3669.02. This work was in part supported by NPRP grant #8-086-1-017 from the Qatar National Research Fund.

## Author contributions

B.C., S.W.B., and Y.H. contributed equally to this work. E.H.S. and S.D.W. supervised the project. B.C., S.W.B., and Y.H. designed the experiment. B.C. carried out the perovskite device fabrications and characterizations. S.W.B. and B.S. carried out CQD synthesis and device characterizations. E.A., M.D.B., T.G.A., and E.V.K. fabricated silicon bottom cells and performed tandem measurements. A.P., Z.H., and M.W. carried out spectroscopic measurements. Y.W. performed XRD measurements. E.H.J., F.G.D.A., M.I.S., and S.H. contributed to data analysis and interpretation. B.C. and S.W.B. wrote the first draft. All authors discussed results, read, and commented on the manuscript.

## Competing interests

The authors declare no competing interests.
