## [Peer Review File · Nature Communications]

Reviewers' comments:

Reviewer #1 (Remarks to the Author):

The primary novel concept in this manuscript is the process by which a thick metal halide perovskite film is formed by first spinning the precursor solution at a low spinning rate and then increasing the spinning rate just before the anti-solvent treatment. In this way is possible to both have a thick film and rapidly falling off the anti-solvent. There are also exciting advances in making a low bandgap quantum dot solar cell more efficient. I think these discoveries merit publication in Nature Communications, but would like to see the authors address a number of concerns expressed below.

The authors report 28% power conversion efficiency for a four-terminal perovskite-Si tandem and 24 % efficiency for a perovskite-quantum dot solar cell tandem. At first these numbers seem to be extraordinary. However, the numbers are less impressive when one realizes how small the solar cells are. There is a graph in this paper showing record efficiencies. For many of those data points, the solar cell was a full square centimeter in size. The supplemental section says that the size of the quantum dot solar cell was 0.049 cm^2 . The perovskite cell had an area of 0.053 cm^2 .

The "4T perovskite-Si" tandems is not really a tandem at all. The authors put a much larger perovskite solar cell stack on top of the Si cell and used the stack as a filter while measuring the Si cell efficiency. They then used the small perovskite cell to get an efficiency for the perovskite cell. Now that prototypes of 4T tandems have been demonstrated repeatedly by several research teams, I think the time has come for the community to stop obtaining numbers this way. It's time to start building real 4T tandems with matched areas. A recently reviewed a manuscript that recommends a proper protocol for measuring efficiencies of 4T tandems.

I have no problem with the authors reporting the numbers for small cells, but I would also like to see them report the efficiency for 1 cm^2 device. They should let people know much the efficiency drops because of the additional series resistance in the electrode. Practical solar panels cannot be made with solar cells that are only 2 mm wide.

It is not true that "The 4T tandem arrangement offers a higher theoretical PCE." The theoretical limit for 2T and 4T tandems are the same. In the theoretical limit, the bandgaps for the 2T tandem would be the ideal ones.

On page 3 the authors state "When we increased perovskite precursor solution concentration, crystallization is less controllable before anti-solvent dripping because of the high supersaturation of precursor solution and fast perovskite reaction rate during subsequent thermal annealing^{33,34}. As a result, perovskite films form with a rough and wrinkled surface (Figure 1a-c), in line with previous reports^{23,24}." They have not accurately summarized the explanation provided in the papers they cite. The wrinkling occurs because compressive stresses arise during the film formation process. I do not agree that roughness arises because it is hard to control the crystallization. More precise wording is needed.

The authors determined grain boundary size by simply looking at scanning electron microscopy images. Many people have asserted that there are actually multiple grains between the lines that are visible in the micrographs. The authors should make a comment on this subject. They might use words such as "apparent grain size." A nice paper that was presented recently on this subject is [10.1016/j.joule.2019.09.001](https://doi.org/10.1016/j.joule.2019.09.001).

The authors should explain why zirconium-doped In_2O_3 is superior to ITO.

Reviewer #2 (Remarks to the Author):

The authors fabricate 1.63 eV perovskite solar cells with thick absorber layers that also have smooth morphology. This results in 19% power conversion efficiency (PCE) for a semi-transparent perovskite solar cell and in turn enabling perovskite-silicon tandem solar cells with a PCE of 28.2%.

While the reported PCE is very impressive, the major conclusions of the work have been reported previously.

Although the demonstrated boosted-solvent-extraction technique creates smooth and thick perovskite films, addition of the Lewis base is key to the electronic performance of the layer. However, the Lewis acid-base adduct approach to improve the properties of the perovskite has already been demonstrated before (ref 34).

The authors report reduced trap densities in the perovskite thus enabling longer diffusion lengths (2.3 μm). However, there is very little data presented in this regard to support these claims.

I am afraid this manuscript does not provide the sort of significant conceptual advance nor represent a sufficiently striking advance to justify publication in Nature Communications.

For these reasons, I do not believe that the manuscript meets the requirements for publication in Nature Communications.

Reviewer #3 (Remarks to the Author):

In this manuscript, the authors reported a four-terminal tandem perovskite-silicon solar cell with a record power conversion efficiency exceeding 28%. The result was enabled by the combination of a thicker perovskite layer, which the authors achieved by modifying the standard spinning and antisolvent procedures, and a defect passivation additive that results in a larger electron diffusion length.

I found the work interesting but not sufficiently novel for publication. Besides a well-written story, this work is combining two optimizations of the perovskite film deposition and then applying them in a four-terminal tandem concept. Therefore I see it more as an incremental than a step improvement in the field.

In conclusion, I cannot recommend publication since of lack of novelty.

Reviewers' comments:

Reviewer #1 (Remarks to the Author):

The primary novel concept in this manuscript is the process by which a thick metal halide perovskite film is formed by first spinning the precursor solution at a low spinning rate and then increasing the spinning rate just before the anti-solvent treatment. In this way is possible to both have a thick film and rapidly falling off the anti-solvent. There are also exciting advances in making a low bandgap quantum dot solar cell more efficient. I think these discoveries merit publication in Nature Communications, but would like to see the authors address a number of concerns expressed below.

We thank the reviewer for the much-valued feedback below on how to increase the impact of the work.

The authors report 28% power conversion efficiency for a four-terminal perovskite-Si tandem and 24 % efficiency for a perovskite-quantum dot solar cell tandem. At first these numbers seem to be extraordinary. However, the numbers are less impressive when one realizes how small the solar cells are. There is a graph in this paper showing record efficiencies. For many of those data points, the solar cell was a full square centimeter in size. The supplemental section says that the size of the quantum dot solar cell was 0.049 cm². The perovskite cell had an area of 0.053 cm².

We have added the device active area into the efficiency summary in Table S1.

Type	PCE (%)	PSC cell area (cm ²)	PSC cell filter area (cm ²)	Reference
4T	28.2	0.049	4	This work
	25.7	1.0	4	This work
	17	0.39	0.39	Energy Environ. Sci. 8, 956–963 (2015)
	19.8	0.0919	0.0919	Science 351, 151–155 (2016)
	21.4	0.09	0.09	ACS Appl. Mater. Interfaces 10, 35016–35024 (2018)
	25.2	0.25	0.25	ACS Energy Lett. 1, 474–480 (2016)
	25.3	0.13	4	Energy Environ. Sci. 11, 1489–1498 (2018)
	25.5	0.09	0.09	ACS Appl. Mater. Interfaces (2019)

	26.1	NA	NA	7th WCPEC 3575–3577 (IEEE, 2018)
	26.2	0.16	6.25	Advanced Functional Materials 29, 1901741 (2019)
	26.4	0.16	4.84	Advanced Energy Materials 7, 1700228 (2017)
	26.9	0.104	0.104	J. Mater. Chem. A 6, 3583–3592 (2018)
	27.1	0.13	0.13	ACS Energy Lett. 4, 259–264 (2019)
2T	23.6	1	NA	Nature Energy 2, 17009 (2017)
	25	1	NA	ACS Energy Lett. 3, 2173–2180 (2018)
	25.2	1.42	NA	Nature Mater 17, 820–826 (2018)

The “4T perovskite-Si” tandems is not really a tandem at all. The authors put a much larger perovskite solar cell stack on top of the Si cell and used the stack as a filter while measuring the Si cell efficiency. They then used the small perovskite cell to get an efficiency for the perovskite cell. Now that prototypes of 4T tandems have been demonstrated repeatedly by several research teams, I think the time has come for the community to stop obtaining numbers this way. It’s time to start building real 4T tandems with matched areas. A recently reviewed a manuscript that recommends a proper protocol for measuring efficiencies of 4T tandems.

I have no problem with the authors reporting the numbers for small cells, but I would also like to see them report the efficiency for 1 cm² device. They should let people know much the efficiency drops because of the additional series resistance in the electrode. Practical solar panels cannot be made with solar cells that are only 2 mm wide.

We have fabricated semi-transparent perovskite front cells with active areas of 1.95 cm².

We measured the device with masks of 0.49 cm² (17.3% PCE), 1 cm² (16.5% PCE), and 1.68 cm² (15.9% PCE), respectively. As seen in the table below, the FF decreases as the device area increases, which leads to drops in PCEs. The increased series resistance arises due to the enlarged area of the transparent conductive oxide (TCO) electrode. We now include a comment in the revised paper that, for devices exceeding 1 cm², it will in future become important to incorporate metal fingers/busbars.

On page 10, we have added: “We also fabricated 1 cm² size tandem, which yields a PCE of 25.7%. This is limited by fill factor due to increased series resistance from TCEs in the semi-transparent PSC (Figure S10), and it will in future become important to incorporate metal fingers/busbars.”

Cells	V _{oc} (V)	J _{sc} (mA/cm ²)	FF (%)	PCE (%)	Stabilized (%)
Semi-transparent PSC					
(0.049 cm ²)	1.12	22.3	77.7	19.4	19.0
(1 cm ²)	1.10	22.4	66.9	16.5	16.5
Filtered CQD	0.62	12.2	66	5.0	5.0
PSC-CQD 4T					
(0.049 cm ²)				24.4	24.0
Filtered SHJ	0.70	17.2	76.6	9.2	9.2
PSC-SHJ 4T					
(0.049 cm ²)				28.6	28.2
(1 cm ²)				25.7	25.7

In the supplementary information, we added Figure S10.

Figure S10 | a, JV curves and b, stabilized power output of 1.95 cm² devices with various mask sizes. Inset is the image of the corresponding device.

It is not true that “The 4T tandem arrangement offers a higher theoretical PCE.” The theoretical limit for 2T and 4T tandems are the same. In the theoretical limit, the bandgaps for the 2T tandem would be the ideal ones.

On page 1, we have revised the statement to read: “The 4T tandem arrangement offers a broader bandgap selection window for its constituent cells.”

On page 3 the authors state “When we increased perovskite precursor solution concentration, crystallization is less controllable before anti-solvent dripping because of the high supersaturation of precursor solution and fast perovskite reaction rate during subsequent thermal annealing^{33,34}. As a result, perovskite films form with a rough and wrinkled surface (Figure 1a-c), in line with previous reports^{23,24}.” They have not accurately summarized the explanation provided in the papers they cite. The wrinkling occurs because compressive stresses arise during the film formation process. I do not agree that roughness arises because it is hard to control the crystallization. More precise wording is needed.

On page 3, we have revised the sentence: “When we increased perovskite precursor solution concentration, the transition from intermediate states to the perovskite phase is fast because of the high supersaturation of precursor solution and rapid perovskite reaction rate during subsequent thermal annealing^{33,34}. However, such a quick intermediate to perovskite phase transition imposes compressive stress because of sudden volume change during the process, resulting in films with a rough and wrinkled surface^{23,24}, as shown in Figure 1a-c.”

The authors determined grain boundary size by simply looking at scanning electron microscopy images. Many people have asserted that there are actually multiple grains between the lines that are visible in the micrographs. The authors should make a comment on this subject. They might use words such as “apparent grain size.” A nice paper that was presented recently on this subject is 10.1016/j.joule.2019.09.001.

We agree with the reviewer that “apparent grain size” is a more appropriate term. We have now included the relevant paper and write apparent grain size in the revised manuscript.

On page 7, we now write: “Figure 3e shows the statistical distribution of the apparent lateral grain size from SEM of perovskite films fabricated under different conditions. Urea-treated 700 nm thick perovskite

films exhibit a larger apparent grain size (700 nm-Large), averaging 1.3 μm compared to 0.6 μm without any additive (700 nm-Small).”

The authors should explain why zirconium-doped In_2O_3 is superior to ITO.

On page 8, we now write: “One of the key enablers of high NIR transmittance is replacing commercial ITO with previously-developed highly-conductive Zr-doped In_2O_3 (IZrO) TCOs, whose parasitic free carrier absorption is suppressed, for a given free carrier density, by virtue of its enhanced carrier mobility⁴⁴.”

Figure S6 | Optical transmittance of the semi-transparent perovskite devices with commercial ITO substrate vs. Zr doped In_2O_3 (IZrO) substrate.

Reviewer #2 (Remarks to the Author):

The authors fabricate 1.63 eV perovskite solar cells with thick absorber layers that also have smooth morphology. This results in 19% power conversion efficiency (PCE) for a semi-transparent perovskite solar cell and in turn enabling perovskite-silicon tandem solar cells with a PCE of 28.2%.

While the reported PCE is very impressive, the major conclusions of the work have been reported previously.

Although the demonstrates boosted-solvent-extraction technique creates smooth and thick perovskite films, addition of the Lewis base is key to the electronic performance of the layer. However, the Lewis acid-base adduct approach to improve the properties of the perovskite has already been demonstrated before (ref 34).

Semi-transparent perovskite top cells in tandem suffer from inadequate absorption of above-bandgap photons, which is due primarily to the lack of effective routes to increase the perovskite thickness while retaining long carrier diffusion length.

We show herein that increasing precursor solution concentration compromises device performance because of rough surface morphology. We sought therefore to develop a novel boosted-solvent-extraction (BSE) technique to enable high J_{sc} and PCE with negligible hysteresis.

This report is the first to demonstrate that the combination of optically thick perovskite film and the Lewis acid-base adduct approach can benefit perovskite tandems.

The authors report reduced trap densities in the perovskite thus enabling longer diffusion lengths (2.3 μm). However, there is very little data presented in this regard to support these claims.

We now provide a new suite of electrical characterization studies, including transient photovoltage (TPV), photocurrent (TPC) and space-charge limited current (SCLC), to compare quantitatively the trap density for passivated vs. control films.

These new studies supplement, and accord with, the findings from time-resolved photoluminescence (TRPL) of Figure 3,

On page 7, we have added the following figures and discussion:

Figure 4 | Effects of grain size on carrier transport and defects density in thick perovskite. **a**, Non-quenched, **b**, Spiro-OMeTAD and **c**, PCBM-quenched time-resolved photoluminescence spectroscopy measurements on perovskite films. **d**, Transient photocurrent, **e**, Photovoltage decay and **f**, Space charge limited current of 700 nm perovskite devices with various grain sizes.

“Fast transient photocurrent (TPC) decay confirms the improvement of charge collection in the treated device (Figure 4d). Fewer grain boundaries in the vertical direction reduce carrier recombination, as evidenced by the longer transient photovoltage (TPV) lifetime (Figure 4e) and smaller trap filling voltage (V_{TFL}) from space charge limited current (SCLC) measurement in electron-only devices (ITO/SnO₂/Perovskite/PCBM/Ag) devices (Figure 4f), resulting in a V_{oc} increase of 20 meV.”

I am afraid this manuscript does not provide the sort of significant conceptual advance nor represent a sufficiently striking advance to justify publication in Nature Communications.

Tandem photovoltaics that involve perovskite cells offer a pathway to increased power conversion efficiencies (PCEs). The perovskite top cells in tandems suffer from inadequate absorption of above-bandgap photons; and this has so far kept published tandem performance below 28% PCE.

In this manuscript, we sought to advance perovskite-based tandem solar cells by a general design strategy. Its implementation enables efficient and transparent perovskite top cells, spectrum-tailored bottom cells for tandem applications. Specifically, we devised a novel fabrication routine to fabricate optically thick perovskite film with long electron diffusion length, and a new organic/colloidal quantum dot (CQD) hybrid for enhanced NIR spectral response.

As a result, we demonstrated a record four-terminal (4T) perovskite/silicon tandem PCE of 28.2%, and perovskite/CQD tandem PCE of 24%.

For these reasons, I do not believe that the manuscript meets the requirements for publication in Nature Communications.

Reviewer #3 (Remarks to the Author):

In this manuscript, the authors reported a four terminals tandem perovskite-silicon solar cell with a record power conversion efficiency exceeding 28%. The result was enabled by the combination of a thicker perovskite layer, which the authors achieved modifying the standard spinning and antisolvent procedures, and a defect passivation additive that result in a larger electron diffusion length.

I found the work interesting but not sufficiently novel for publication. Besides a well-written story, this work is combining two optimisations of the perovskite film deposition and then applying them in a four-terminal tandem concept. Therefore I see it more as an incremental than a step improvement in the field.

In conclusion, I cannot recommend publication since lack of novelty.

Tandem photovoltaics that involve perovskite cells offer a pathway to increased power conversion efficiencies (PCEs). The perovskite top cells in tandems suffer from inadequate absorption of above-bandgap photons; and this has so far kept published tandem performance below 28% PCE.

We show herein that increasing precursor solution concentration compromises device performance because of rough surface morphology. We sought therefore to develop a novel boosted-solvent-extraction (BSE) technique to enable high J_{sc} and PCE with negligible hysteresis.

This report is the first to demonstrate that the combination of optically thick perovskite film and the Lewis acid-base adduct approach can benefit perovskite tandems.

In this manuscript, we sought to advance perovskite-based tandem solar cells by a general design strategy. Its implementation enables efficient and transparent perovskite top cells, spectrum-tailored bottom cells for tandem applications. Specifically, we devised a novel fabrication routine to fabricate optically thick perovskite film with long electron diffusion length, and a new organic/colloidal quantum dot (CQD) hybrid for enhanced NIR spectral response.

As a result, we demonstrate a record four-terminal (4T) perovskite/silicon tandem PCE of 28.2%, and perovskite/CQD tandem PCE of 24%.

REVIEWERS' COMMENTS:

Reviewer #1 (Remarks to the Author):

I am satisfied with the responses to the reviewer comments and am able to recommend publishing the manuscript in its current form.

Reviewer #2 (Remarks to the Author):

The authors have provided a new suite of electrical characterisation and measurements. Additionally, they have demonstrated semi-transparent perovskite solar cells (and tandem solar cells) that are significantly larger than the ones reported previously while exhibiting relatively high efficiency.

The authors have significantly improved the article in the revised manuscript and I believe this revised article merits a publication in Nature Communications.

Therefore, I recommend the publication of the revised manuscript in Nature Communications.

REVIEWERS' COMMENTS:

Reviewer #1 (Remarks to the Author):

I am satisfied with the responses to the reviewer comments and am able to recommend publishing the manuscript in its current form.

We thank the referee for a constructive review process.

Reviewer #2 (Remarks to the Author):

The authors have provided a new suite of electrical characterisation and measurements. Additionally, they have demonstrated semi-transparent perovskite solar cells (and tandem solar cells) that are significantly larger than the ones reported previously while exhibiting relatively high efficiency.

The authors have significantly improved the article in the revised manuscript and I believe this revised article merits a publication in Nature Communications.

Therefore, I recommend the publication of the revised manuscript in Nature Communications.

We thank the referee for a constructive review process.